# Understanding climate-sensitive diseases in Bangladesh using systematic review and government data repository

Md Iqbal Kabir[1,2]*, Dewan Mashrur Hossain[1], Md. Toufiq Hassan Shawon[3], Md. Mostaured Ali Khan[4], Md Saiful Islam[1], As Saba Hossain[1], Md Nuruzzaman Khan[5,6]

1 Climate Change and Health Promotion Unit (CCHPU), Health Services Division, Ministry of Health and Family Welfare (MoHFW), Dhaka, Bangladesh, 2 Department of Disaster Science and Climate Resilience, University of Dhaka, Dhaka, Bangladesh, 3 Management Information System (MIS), Directorate General of Health Services, Ministry of Health and Family Welfare, Mohakhali, Dhaka, Bangladesh, 4 Maternal and Child Health Division, icddr,b, Mohakhali, Dhaka, Bangladesh, 5 Department of Population Science, Jatiya Kabi Kazi Nazrul Islam University, Trishal, Mymensingh, Bangladesh, 6 Nossal Institute for Global Health, Melbourne School of Population and Global Health, The University of Melbourne, Australia

* Iqbalkabirdr@gmail.com

## Abstract

### Background

Understanding the effects of climate change on health outcomes is crucial for effective policy formulation and intervention strategies. However, in Low- and Middle-Income Countries, like Bangladesh, the true extent of these effects remains unexplored due to data scarcity. This study aims to assess available evidence on climate change-related health outcomes in Bangladesh, to compare it with actual national occurrences, and to explore challenges related to climate change and health data.

### Methods

We first conducted a systematic review to summarize the climate-sensitive diseases examined in existing literature in Bangladesh. The review results were then compared with over 2.8 million samples from the government's data repository, representing reported cases of climate-sensitive diseases during 2017-2022. This comparison aimed to identify discrepancies between the diseases currently occurring in Bangladesh related to climate change and available knowledge through existing research. Additionally, we also explored the limitations of the data recorded in the government data repository.

### Results

The available literature in Bangladesh reports only a few specific climate-sensitive diseases, including Diarrhea, Dengue, Cholera, Malaria, Pneumonia, Cardiovascular Diseases, Hypertension, Urinary-Tract Infections, and Malnutrition, which were also considered in few studies. This represents a segment of the total 510 reported climate-sensitive diseases in Bangladesh, of which 143 diseases were responsible for 90.66% of the total occurrences. The most common forms of diseases were diarrhea and gastroenteritis of

**Data availability statement:** The data supporting the findings of this study are accessible through MoHFW of Bangladesh but are not publicly available. Researchers interested in accessing the dataset can do so by submitting a research proposal to MoHFW, similar to the process we followed to obtain the dataset for this study. The dataset can be accessed at http://www.mohfw.gov.bd by submitting a formal application through the same link.

**Funding:** The author(s) received no specific funding for this work.;

**Competing interests:** The authors have declared that no competing interests exist.

presumed infectious (28.51%), pneumonia (18.88%), anxiety disorders, panic disorders, generalized anxiety disorders (13.2%), and others (13.15%). Additionally, Urinary-Tract infections (7.87%), cholera (3.03%), and typhoid fever (3.27%) were other frequently reported climate-sensitive diseases. We also explored several challenges related to available data in the government repository, which include inadequate collection of patients' comprehensive socio-demographic information and the absence of a unique patient identifier.

## Conclusion

The findings underscore the urgent need to tackle data challenges in understanding climate-sensitive diseases in Bangladesh. Policies and programs are required to prioritize the digitalization of the healthcare system and implement a unique patient identification number to facilitate accurate tracking and analysis of health data.

Climate Change, including rising temperature and extreme weather events like cyclone and floods, poses a significant global health threat [1]. The World Health Organization estimates climate change already causes at least 150,000 deaths annually at the global level, and that number is projected to double by 2030. Beside these other impact of climate change are far-reaching, leading to forced displacement, malnutrition and increased incidence of diseases such as dengue, diarrhea, and pneumonia [2]. Additionally, climate change has established links to mental health issues, like anxiety and depression [3]. The effects are particularly severe in Low- and Middle-Income Countries (LMICs) due to limited resources and inadequate infrastructure for coping with erratic weather and disasters [4]. We undertook a comprehensive mixed-method study, incorporating a systematic review of existing studies conducted in Bangladesh, along with an analysis of government data repository. A detailed description of each component is presented below.

## Background

Bangladesh, a LMIC located in South Asia, ranks seventh among countries most vulnerable to climate change due to its vast coastal area, high population density and high poverty rate [5]. There is an estimate that one in every seven people in Bangladesh will be displaced by 2050 because of climate change, particularly due to sea level rise. This would results in approximately loss of 11% of the country's total land area and migration of up to 18 million people [6]. These long-term consequences compounded the regular occurrence of adverse climate events, for instances, floods, cyclone, flash floods and landslides that affect Bangladesh almost every year [7].

Adverse climate change events pose serious risks to disease outbreaks in Bangladesh. Sixty percent of global cyclone-related deaths in the past 20 years occurred there, either because of casualty due to cyclone and/or post-cyclonic adverse health outcomes [6]. Moreover, at least 117 million population will be at risk of facing malaria by 2070, potentially rising to 147 million under high emission situation [8]. Other climate sensitive diseases, including dengue, chikungunya, kala-azar, and cholera, are increasingly prevalent in Bangladesh [5,8]. These pose risk to achieving the Universal Health Coverage (UHC), a key focus of Bangladesh's Sustainable Development Goals. Inadequate funding, infrastructure, resources, logistic, and services in the healthcare system further compound the risk of climate change impact [9].

The Government of Bangladesh formulated a Climate Change Strategy and Action Plan in 2008, later updated in 2009, to tackle climate-sensitive diseases [10]. The plan aims to comprehensively assess the prevalence of climate-sensitive diseases across the country, considering geographical variations, and implement targeted policies and programs accordingly. However, this effort is impeded by the predominant focus on specific diseases such as diarrhea, dengue, cholera, malaria, pneumonia, cardiovascular diseases, hypertension, urinary-tract infections, and malnutrition in existing evidence [11,12]. Establishing a surveillance system to monitor both existing and emerging climate-related diseases and strengthening health system resilience for the future are top priorities. Nevertheless, the effectiveness of these initiatives is hampered primarily by the lack of pertinent data. Currently available data mostly originates from small-scale regional studies, which often focus on a few specific outcomes and yield conflicting findings [13–16]. Although the government-initiated data collection and utilization efforts through District Health Information System 2 (DHIS 2) in 2009, the usability and coverage of this data remain largely unexplored. As a result of these complexities, there are limited knowledge regarding the climate change-related health outcomes in Bangladesh as compared to the global perspective, and the true extent of climate-sensitive diseases in the country remains unknown. This hampers the development of effective policies and programs to address the health impacts of climate change. To overcome this limitation, we conducted a mixed-methods study to explore the climate-sensitive diseases reported in existing literature in Bangladesh and compared them with the reported scenarios of relevant data in the government data repository. Furthermore, we examined the challenges associated with reporting climate change and related diseases data in the government data repository.

## Methods

### Systematic review

We performed a systematic review and adhered to the Preferred Reporting Items for Systematic Reviews and Meta-Analyses (PRISMA) guidelines for meta-analyses of observational studies. We included studies that were relevant and accessible on the impact of climate change on health.

### Searches

We conducted a systematic literature search initially in December 2022 and later updated it in July 2023 to include any additional papers published since the initial search. Total of six databases (Medline, Embase, Maternity and Infant Care, Scopus, PsycINFO, and CINHL) were searched. Additional searches were conducted in the Google and Google Scholar and reference lists of included papers. The full search strategy and results are presented in the Supplementary Table 1-6 of the S1 File. The search strategy was developed including the keywords related to climate change related terms combined using the Bolen operator "OR". These include *climate change* OR *environmental disaster* OR *environmental degradation* OR *environmental issues* OR *adaptation* or *vulnerable community* OR *vulnerabilities*. The study focused on *Bangladesh* as the setting. To combine the search results, we used another Boolean operator, "AND". We did not impose any restrictions on the diseases related to climate change to encompass all climate-sensitive diseases recorded in the existing evidence.

### Study selection

Two authors (Khan MN and Islam MS) performed a comprehensive review of all articles, adhering to the inclusion and exclusion criteria outlined in Table 1. Initially, they conducted

**Table 1. Inclusion and exclusion criteria used to select the study to explore the effects of climate change on health outcomes in Bangladesh.**

| Characteristics | Inclusion criteria | Exclusion criteria |
|---|---|---|
| Language | Both Bengali and English | None |
| Study design | All study design | None |
| Place of studies | Bangladesh | Other than Bangladesh |
| Publication status | Published from January 2000 to July 2023, aligning with the beginning of the Millennium Development Goals in January 2000. | Published before January 2000 |
| Paper | Peer reviewed published journal articles | Conference presentation, editorials, letters to the editor, commentaries, review paper, and symposium proceedings |
| Outcome | Any health-related outcomes | Other than health-related outcomes |

a screening of titles and abstracts, and articles selected during this stage underwent full-text review. Any disagreements were resolved through discussions between the two authors, and involvement of senior author (Kabir MI) was sought when necessary. Online platforms such as COVIDENCE, EndNote, and Zoom Online meetings were utilized for conducting this review.

## Data extraction

A data extraction template was developed, tested, and refined prior to final data compilation. Two authors (Khan MN and Islam MS) independently extracted relevant information from the selected studies, including authors' names, study design, sample size, study setting, and specific categories of climate-sensitive diseases. Any discrepancies between the data collectors were resolved through discussions, with involvement from the senior author (Kabir MI) if needed.

## Quality assessment of included studies

The quality assessment of the included studies was conducted using the modified Newcastle-Ottawa Scale [17]. The scale encompassed specific criteria for cross-sectional (n = 26), case-control (n = 2), cohort studies (n = 2), qualitative studies (n = 3), and randomized control trial (n = 1). Authors reviewed the articles and marked an "*" for each criterion met. The scores were then summed up and categorized into three groups: high quality study (if the study achieved over 75% of the total allocated score), moderate quality (if the study achieved 50 to 74% of the total allocated score), and low quality (if the study achieved below 50% of the total allocated score). The majority of the studies were of moderate quality (n = 27), followed by low quality (n = 6) and only one high-quality study (n = 1) (Supplementary table 7-11 of the S1 File).

## Study variables

All adverse events related to climate change were considered as study variables. These events include temperature, rainfalls, floods, droughts, and cyclones. A full list of climatic events can be found in the search strategy presented in the Supplementary table 1-6 of the S1 File.

## Outcome variables

Several adverse health outcomes were considered as outcome variables. The full list of outcome variables is available in the fourth column of the Table 1.

### Exploration of repository health data from governmental database

The government of Bangladesh in 2009 started recording real-time healthcare service utilization data for every patient admitted in to the divisional, district, and upazila level government hospitals through DHIS 2 platform. The data was recorded under the supervision of the Ministry of Health and Welfare of Bangladesh. They have a separate unit called the Management Information System (MIS) to record this information along with other relevant data. They also conducted regular monitoring visits to ensure data accuracy. At the field level, they have expert personnel to record data. The data collection is currently being conducted at 516 healthcare facilities across the country, selected based on their infrastructural capabilities to record real-time data. Both aggregated and individual-level data were recorded, but for our analysis, we focused solely on the individual-level data and climate sensitive diseases as per International Classification of Diseases-10 (ICD-10). The data was collected by authorized personnel, including statisticians and medical staff, using a web-based platform during the provision of treatment to patients. The information was automatically stored in the National Health Information System database of Bangladesh.

### Analysis

The heterogeneity of the included studies through systematic review precluded quantitative analysis of the data. We therefore used narrative synthesis to summarize the findings of all retrieved studies. There were no missing data in the included papers. We used descriptive statistics to explore the quantitative data recorded in the DHIS 2. Stata version 15.1/MP (StataCorp, College Station, Texas USA) was used for statistical analysis.

### Ethics Approval

We conducted an analysis of deidentified secondary data obtained from the Ministry of Health and Family Welfare (MoHFW) of Bangladesh, along with a systematic review of published papers. Since both datasets were deidentified, ethical approval was not required.

## Results

### Study selection

A total of 1420 papers were initially identified through a comprehensive search across six databases, supplemented by additional identification of 11 papers via Google, Google Scholar, and a review of reference lists from the selected articles. After removal of duplicate entries, a refined list of 1367 unique articles was obtained (Supporting Information File 2). Upon screening of titles and abstracts based on predefined inclusion and exclusion criteria, 886 articles were excluded, as depicted in Fig 1. This left 315 articles for full-text review, among them 281 articles were excluded because of no outcome (n = 245) and wrong study design (n = 36). Finally, 34 studies were included in qualitative synthesis.

An abridged representation of all the papers included in the current research is available in Supplementary Table 12 of the S1 File. A summary of the key findings derived from these papers is presented in Table 2. A majority of the selected study were cross-sectional (n = 26), followed by case-control study (n = 2), cohort study (n = 2), and randomized control trial (n = 1). Moreover, three of the included studies were qualitative. Majority of these studies conducted at the regional level (n = 29). A substantial portion of these studies considered climate induced communicable diseases, including diarrhea (n = 15), dengue (n = 4), cholera (n = 5), malaria (n = 4), and pneumonia (n = 4). non-communicable diseases, including cardiovascular

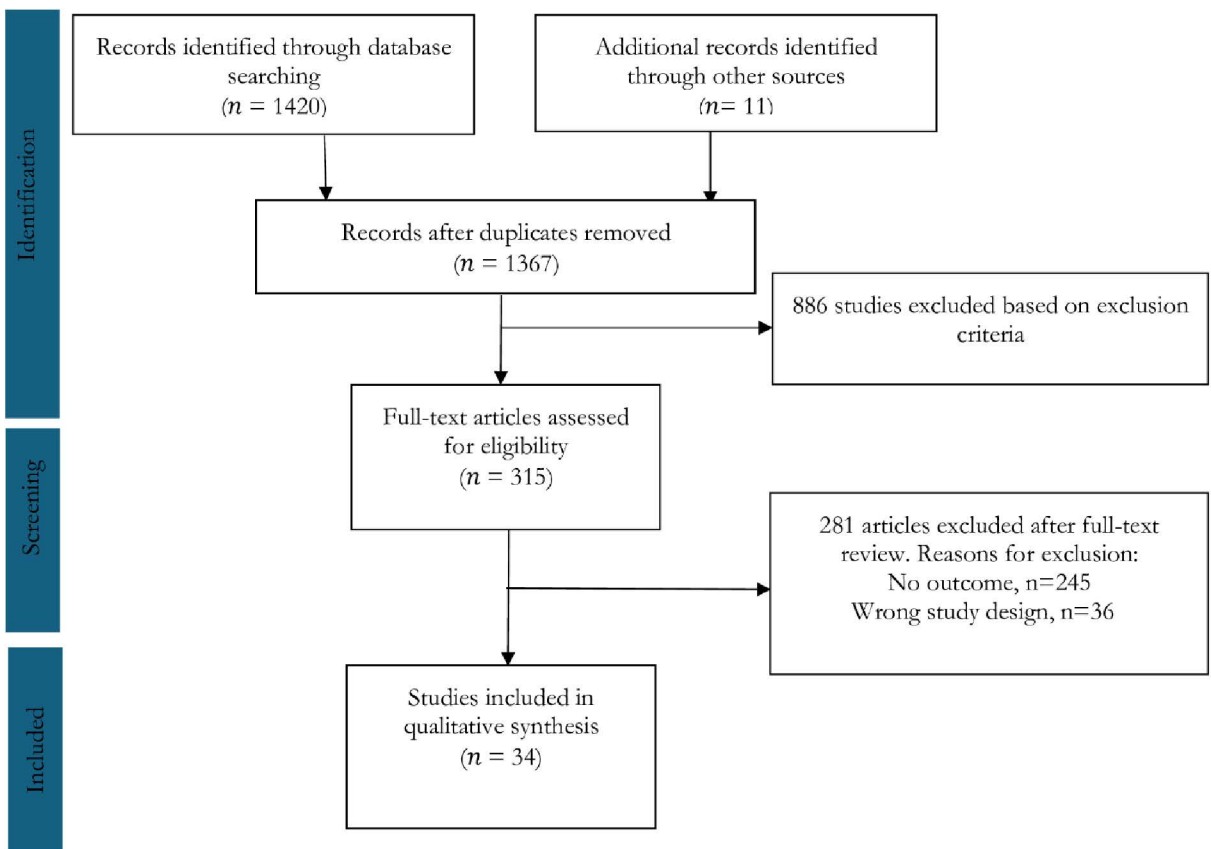

**Fig 1. PRISMA flow diagram illustrating the study selection process covering effects of climate change on disease in Bangladesh.**

diseases (n = 3), hypertension (n = 4), urinary-tract infections (n = 2), and malnutrition (n = 2) were also considered in few studies.

### Exploration from quantitative data extracted from the government data repository

**Background characteristics of the respondents.** The climate sensitive diseases data were found to be recorded 516 healthcare facilities and we included all in the analysis. The district (secondary administrative unit of Bangladesh) wise distribution of these healthcare facilities along with risk of climate change events is presented in Fig 2. The dataset comprising 2,865,365 records of individuals who reported any form of climate-sensitive diseases within the timeframe spanning from January 2017 to November 2022. The climate-sensitive diseases were classified using the ICD-10, as presented in the Supplementary Table 13 of S1 File. The distribution of samples across different years revealed that the highest number of cases was recorded in 2019, accounting for 22.50% of the total dataset. This was followed by 2022, which accounted for 20.54%, and 2018, with 18.88% of the cases (Table 3).

### Sample characteristics

The distribution of the analyzed sample across socio-demographic characteristics of the respondents is presented in Table 4. We found a higher prevalence of climate-sensitive diseases among female (55.82%) following male (44.16%). Under-five aged children were found

**Table 2. Summary of the existing literature in Bangladesh for the period 2000-2022 covering adverse climate events and climate sensitive diseases.**

| | Author's, Year of publication | Study type | Study location | Outcome |
|---|---|---|---|---|
| Communicable diseases (CDs) | Hu et al. 2014 [18] | Regional | Dhaka | Dengue |
| | Hossain et al. 2019 [19] | National | Bangladesh | Dengue |
| | Sharker et al. 2020 [20] | Regional | Dhaka | Dengue |
| | Lorah et al. 2022 [21] | National | Bangladesh | Cholera |
| | Ishimura et al. 2008 [22] | Regional | Dhaka | Cholera |
| | Rheman et al. 2009 [23] | Regional | Matlab | Cholera |
| | Yunus et al. 2018 [24] | Regional | Matlab | Cholera |
| | Yunus et al. 2014 [25] | Regional | Matlab | Diarrhoea |
| | Mollah et al. 2014 [26] | Regional | Dhaka | Diarrhoea |
| | Mollah et al. 2014 [27] | Regional | Dhaka | Asthma |
| | Grembi et al. 2022 [28] | Regional | Gazipur, Kishoreganj, Mymensingh, Tangail | Diarrhoea |
| | Nguyen et al. 2022 [14] | Regional | Gazipur, Kishoreganj, Mymensingh, Tangail | Diarrhoea |
| | Armstrong et al. 2007 [15] | Regional | Patients visiting (ICDDR, B), Dhaka | Non-Cholera Diarrhoea |
| | Hashizume et al. 2010 [29] | Regional | Rangamati district hospital | Malaria |
| | Adegboye et al. 2020 [30] | Regional | UHC,Rajasthali, Rangamati | Malaria |
| | Tong et al. 2020 [31] | Regional | Matlab | Pneumonia |
| | Ibrahim et al. 2018 [13] | National | Bangladesh | Malaria, Diarrheal Disease, Enteric Fever, Encephalitis, Pneumonia, and Bacterial Meningitis. |
| | Hashizume et al. 2016 [32] | Regional | Mymensingh, Tangail, Gazipur, Pabna, Jamalpur, Khulna, Panchagar, Rajshahi, and Sirajganj. | Kala-Azar |
| | Nurhamim 2020 [33] | National | Bangladesh | Skin Infection, Pneumonia, Respiratory Infection, Mosquito-Borne Illnesses, Hepatitis A Or E Virus Infection. |
| | Rahman et al. 2016 [34] | Regional | Bagerhat, Barguna, Cox's Bazar, Faridpur, Khulna, Satkhira, and Sirajganj | Dengue, Malaria, Diarrhea, and Pneumonia |
| | Ashrafuzzaman and Furini 2019 [35] | Regional | Shyamnagar Upazila | Dysentery, Skin Diseases and Diarrhea |
| | Parr et al. 2019 [36] | Regional | North-western mainland region of Bangladesh | Fever, Diarrhea, Jaundice, Typhoid, Acute Respiratory Infections and Gastrointestinal Diseases |
| | Shi et al. 2022 [37] | Regional | Gaibandha | Skin Diseases and Diarrhea |
| Non-communicable diseases (NCDs) | Rutherford et al. 2016 [38] | Regional | Koyra, (Khulna) | Hypertension, Cardiovascular Diseases, Kidney Diseases, Malnourished |
| | Khan et al. 2019 [11] | Regional | Mathbaria,Zianagar, Mongla | Cardiovascular, Diarrhea, Abdominal pain, Gastric ulcer, Dysentery, Skin Diseases, Typhoid |
| | Chowdhury et al. 2017 [39] | Regional | Dacope, Batiaghata, Paikghaccha | High blood pressure, Hypertension |
| | Khan et al. 2016 [40] | Regional | Dacope, Khulna | Hypertension |
| | Siddique et al. 2016 [41] | Regional | Chakaria | Eclampsia, Hypertension, Cardiovascular Diseases, Cancer |
| Maternal Health Issues | Rashid and Michaud 2000 [42] | Regional | Manikganj, Dhaka | Gota and Chulkani, Perineal Rashes, Cramps and Urinary-Tract Infections, Fever, Diarrhea and Jaundice |
| | Haq A et al. 2021 [43] | National | Bangladesh | Fertility |
| | Dalal et al. 2019 [44] | Regional | Khaliajhuri (Netrakona) | Malnutrition and Anemia, Urinary Tract Infections. |
| Mental Health Issues | Kabir 2018 [16] | Regional | Chattogram, Cox's Bazar, Rangamati, Bandarban,Khagrachhar | Depression, Frustration, and Suicide Tendency |

*(Continued)*

**Table 2.** (Continued)

| | Author's, Year of publication | Study type | Study location | Outcome |
|---|---|---|---|---|
| CDs & NCDs | Baernighausen et al. 2021 [45] | Regional | Bhola slum, Dhaka | Fever, Diarrhoea, Cough, Psychological Trauma, Body Aches |
| | Ashraf and Faruk 2018 [46] | Regional | Dhaka | Diarrhea and Cholera, Sweating, Feeling Thirsty, Discomfort, Headache, Stomach Aches, Prickly-Heat, Getting Easily Irritated, Feeling Sluggish, Weakness and Dehydration, Cold and Fever, Irritation in Skin, Loss of Concentration |

to have the highest incidence of climate-sensitive diseases, comprising 33.13% of all cases. Additionally, respondents aged 5-19 years accounted for 13.93% of cases, followed by those aged 20-29 years (14.89%) and 30-39 years (11.94%). In terms of geographical distribution, the Rajshahi division exhibited the highest occurrence of climate-sensitive diseases at 18.27%, followed by Chattogram at 17.60%, Dhaka at 16.14%, and Khulna at 14.71%.

### Distribution of climate sensitive diseases

We observed a total of 510 cases of climate-sensitive diseases in the quantitative data we analyzed, as indicated in the Supplementary Table 13 of the S1 File. These cases represented nearly 94% of the 540 climate-sensitive diseases summarized in the ICD-10 climate-sensitive diseases mapping. Out of the 510 recorded climate-sensitive diseases, 143 diseases were responsible for 90.66% of the total occurrences. We reclassified these diseases into 14 categories based on their similar types which are presented in Fig 3 and Supplementary Table 14 and 15 of the S1 File. District wise distribution of these are presented in Supplementary Table 16 of the S1 File. Diarrhea and gastroenteritis of presumed infectious origin were the most prevalent climate-sensitive diseases, accounting for 28.51% of the cases (Fig 3). Other significant diseases included various forms of pneumonia (18.88%) and anxiety disorders, panic disorders, generalized anxiety disorders, and others (13.15%). Additionally, urinary tract infections (7.87%), cholera (3.03%), and typhoid fever (3.27%) were frequently reported climate-sensitive diseases.

The distribution of these more prevalent diseases was examined on a yearly basis, and the findings are presented in Table 5. We did not find any specific trend of climate-sensitive diseases, with some years showing a notable increase that subsequently declined in the following years. In general, the prevalence of most of these diseases showed an increase in 2019, except for cholera, which exhibited an increase in 2017.

## Discussion

This study aimed to examine the climate-sensitive diseases documented in the existing literature and to compare them with the government data repository, while also exploring challenges related to recording diseases associated with climate change. Our findings indicate the existing literature focuses on a limited number of climate-sensitive diseases, such as Diarrhea, Dengue, Cholera, Malaria, Pneumonia, Cardiovascular Diseases, Hypertension, Urinary-Tract Infections, and Malnutrition, which have been examined in only a few studies. These diseases constitute a fraction of the 510 reported climate-sensitive diseases, with 143 of them contributing to 90.66% of the total occurrences. Moreover, the government-recorded data have several limitations, posing significant challenges for policymakers and program developers in effectively addressing climate-sensitive diseases. Therefore, there is an urgent need to improve efforts in reporting and documenting all climate-sensitive diseases, along with the development of comprehensive policies and programs to address them effectively.

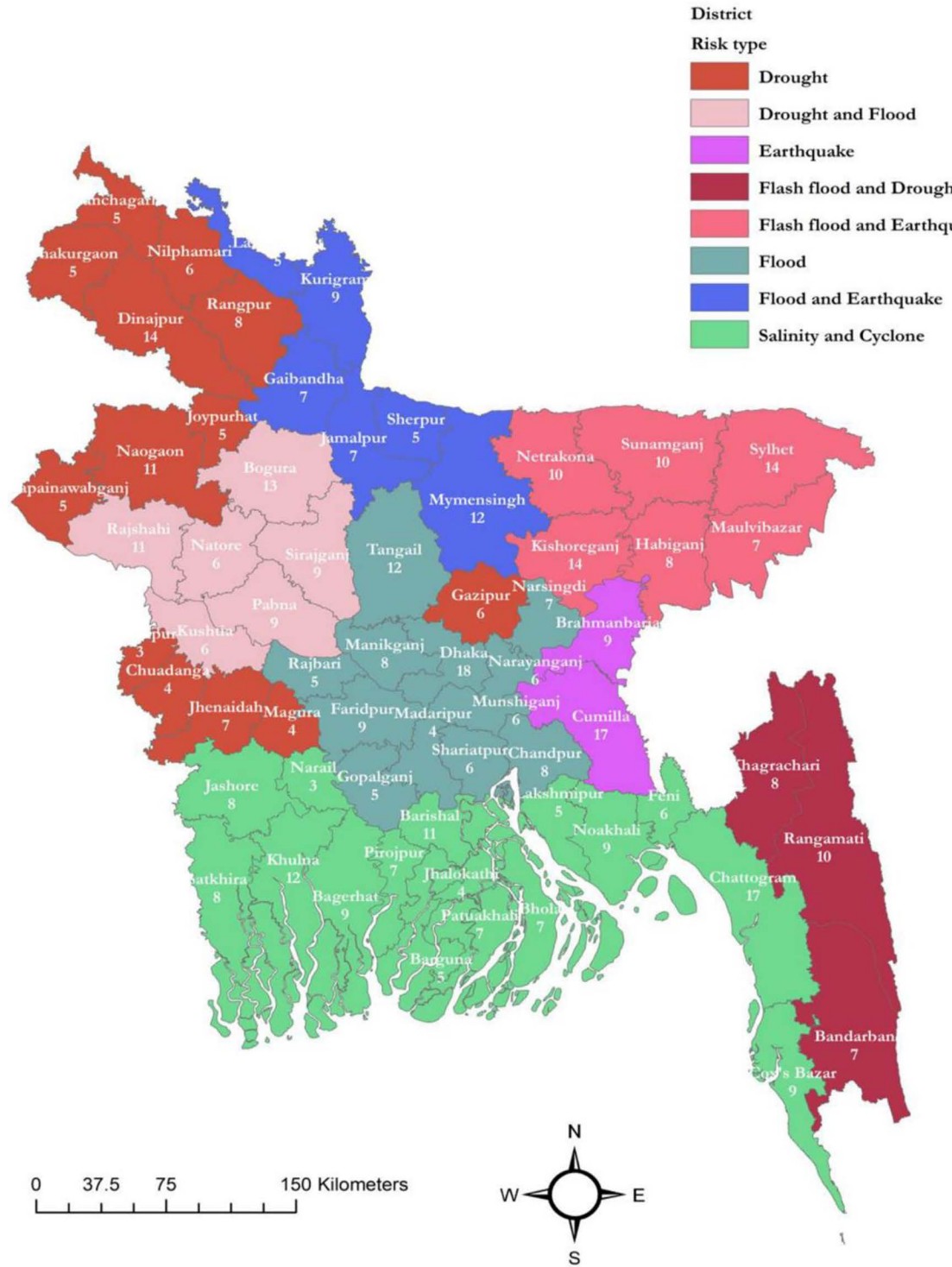

**Fig 2. Distribution of climate vulnerable areas with number of healthcare facilities from where DHIS 2 data were recorded.** (The auhtors created this map using the freely available shapefile from https://data.humdata.org/dataset/cod-ab-bgd. AreGIS 10.1 was used for this purpose. No Third party permission is required to publish it).

**Table 3. Distribution of the data according to the years reported.**

| Year | Number | Percentage |
|------|--------|-----------|
| 2017 | 358897 | 12.53 |
| 2018 | 540928 | 18.88 |
| 2019 | 644597 | 22.50 |
| 2020 | 201699 | 7.04 |
| 2021 | 530626 | 18.52 |
| 2022 | 588618 | 20.54 |

**Table 4. Basic characteristics of the climate sensitive diseases patients.**

| Characteristics | Number | Percentage |
|-----------------|--------|-----------|
| Sex | | |
| Male | 1264534 | 44.16 |
| Female | 1598525 | 55.82 |
| Third gender | 677 | 0.02 |
| **Patient's Age** | | |
| <5 | 836131 | 33.13 |
| 5–19 | 351594 | 13.93 |
| 20–29 | 375742 | 14.89 |
| 30–39 | 301356 | 11.94 |
| 40–49 | 235917 | 9.35 |
| 50–59 | 190663 | 7.56 |
| 60–69 | 140560 | 5.57 |
| 70–79 | 64775 | 2.57 |
| ≥80 | 26676 | 1.06 |
| **Division** | | |
| Barishal | 212755 | 7.43 |
| Chattogram | 504309 | 17.6 |
| Dhaka | 462374 | 16.14 |
| Khulna | 421573 | 14.71 |
| Mymensingh | 182203 | 6.36 |
| Rajshahi | 523542 | 18.27 |
| Rangpur | 312883 | 10.92 |
| Sylhet | 221577 | 7.73 |
| Unrecognised | 24150 | 0.84 |

The existing literature in Bangladesh primarily focuses on the impact of climate change on specific health outcomes. For example, dengue outbreaks are extensively studied as a major climate change-related disease in Bangladesh [18–20]. Weather-related factors like temperature, humidity, and rainfall play a critical role in the proliferation of vectors, viruses, and ecological factors associated with dengue [16,19,37]. While individuals of all age groups are susceptible to the disease, women, children, and the elderly have been identified as more vulnerable populations. Conversely, cholera is commonly observed in children, with heat-waves, rainfall, temperature, and water pH level being reported as underlying factors [16,37]. Childhood diarrheal diseases are also linked to climate change, particularly during flooding, due to the impact on safe water and sanitation [23,39,41].

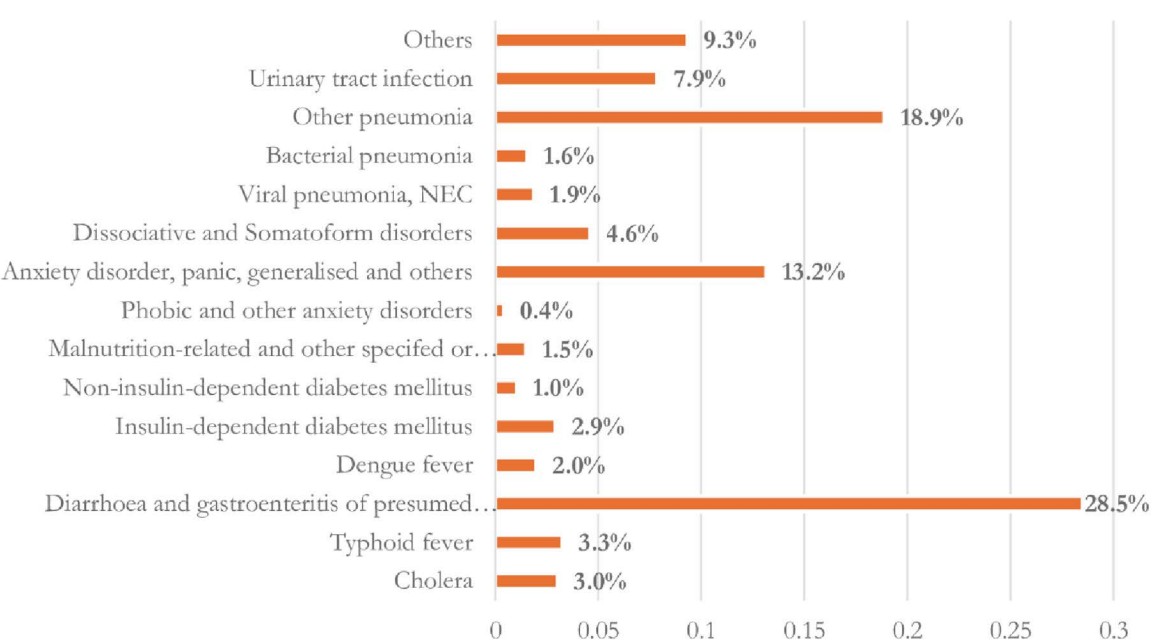

**Fig 3. Major disease related to climate change in Bangladesh over the year 2017-2022.**

**Table 5. Years-wise distribution of most prevalent climate-sensitive diseases in Bangladesh, year 2017-2022.**

| Climate Sensitive Diseases | 2017 | | 2018 | | 2019 | | 2020 | | 2021 | | 2022 | |
|---|---|---|---|---|---|---|---|---|---|---|---|---|
| | n | % | n | % | n | % | n | % | n | % | n | % |
| Cholera | 24499 | 28.21 | 18034 | 20.77 | 12466 | 14.36 | 0 | 0.00 | 16947 | 19.52 | 14894 | 17.15 |
| Typhoid fever | 19732 | 21.07 | 22526 | 24.05 | 23459 | 25.04 | 0 | 0.00 | 12081 | 12.90 | 15870 | 16.94 |
| Diarrhoea and gastroenteritis of presumed infectious origin | 74008 | 9.06 | 180089 | 22.05 | 201528 | 24.67 | 0 | 0.00 | 206373 | 25.26 | 154910 | 18.96 |
| Dengue fever | 414 | 0.72 | 1295 | 2.27 | 26756 | 46.84 | 850 | 1.49 | 3799 | 6.65 | 24009 | 42.03 |
| Insulin-dependent diabetes mellitus | 8506 | 10.15 | 12238 | 14.60 | 14949 | 17.84 | 11003 | 13.13 | 14551 | 17.36 | 22553 | 26.91 |
| Non-insulin-dependent diabetes mellitus | 2290 | 7.70 | 3601 | 12.10 | 5065 | 17.02 | 4541 | 15.26 | 5295 | 17.79 | 8966 | 30.13 |
| Malnutrition-related and other specified or unspecified diabetes mellitus | 4152 | 9.70 | 5907 | 13.80 | 7778 | 18.17 | 6463 | 15.10 | 7422 | 17.34 | 11080 | 25.89 |
| Phobic and other anxiety disorders | 2086 | 16.58 | 2167 | 17.23 | 2741 | 21.79 | 1855 | 14.75 | 1986 | 15.79 | 1744 | 13.86 |
| Anxiety disorder, panic, generalised and others | 47584 | 12.63 | 69952 | 18.56 | 77013 | 20.44 | 47945 | 12.72 | 57882 | 15.36 | 76438 | 20.29 |
| Dissociative and Somatoform disorders | 21297 | 16.13 | 25579 | 19.38 | 26624 | 20.17 | 16857 | 12.77 | 18192 | 13.78 | 23462 | 17.77 |
| Viral pneumonia, NEC | 10229 | 18.97 | 10060 | 18.65 | 11309 | 20.97 | 4042 | 7.49 | 9841 | 18.25 | 8454 | 15.67 |
| Bacterial pneumonia | 7383 | 16.43 | 9631 | 21.43 | 10151 | 22.59 | 2765 | 6.15 | 4887 | 10.87 | 10121 | 22.52 |
| Other pneumonia | 77921 | 14.40 | 89183 | 16.48 | 122484 | 22.64 | 50363 | 9.31 | 95487 | 17.65 | 105646 | 19.52 |
| Urinary tract infection | 25399 | 11.26 | 39244 | 17.40 | 48834 | 21.65 | 28681 | 12.72 | 33117 | 14.69 | 50240 | 22.28 |
| Others | 33397 | 12.48 | 51422 | 19.22 | 53440 | 19.97 | 26334 | 9.84 | 42767 | 15.98 | 60231 | 22.51 |

**Note:** Row percentage was presented in the table

Furthermore, climate-sensitive diseases contribute significantly to the loss of Disability Adjusted Life Years (DALYs, years of life lost due to premature mortality and years lived with disability or illness) among children. Malaria, pneumonia, and malnutrition-related outcomes like stunting, wasting, and underweight have been identified as prominent factors [28,42]. Selected studies have also documented the loss of DALYs related to other adverse

climate-sensitive diseases [34, 35]. In addition to these direct adverse health outcomes, numerous studies establish a link between climate change and an increase in adult health conditions such as high blood pressure, cardiovascular diseases, abdominal pain, gastric ulcers, dysentery, skin diseases, and typhoid, often resulting from water salinity [18,30,32].

Some studies also explore the relationship between climate change and maternal health issues, including menstrual hygiene and the use of maternal healthcare services, as like experience of other LMICs facing adverse effects of climate change [12,42,47]. During floods, the crowded shelter conditions pose challenges for proper menstruation management, particularly among women and adolescent girls [34]. Adverse climate events like cyclones, floods, and droughts reduce the utilization of maternal healthcare services, including antenatal, delivery, and postnatal care, which contributes to an increased risk of pregnancy complications and maternal mortality [23,40,41].

Despite the valuable insights provided by the research on the adverse health effects of climate change in Bangladesh, it is crucial to acknowledge that the available studies only cover a fraction of the total climate-sensitive diseases recorded globally and within the government data repository, as reported in this study. This limitation primarily arises from the lack of reliable and accessible data on climate change and its impact on health [20]. The scarcity of such data hinders the accurate identification and quantification of specific health risks associated with climate change within the country [24]. Furthermore, the research capacity and resources in Bangladesh, as a LMICs, face inherent limitations. Insufficient funding, inadequate infrastructure, and a shortage of skilled researchers pose significant obstacles to conducting comprehensive studies [34]. These constraints can compromise the quality and breadth of research conducted, as well as the ability to gather and analyse data on a larger scale [20]. Moreover, the intricate and multifaceted nature of climate change and its complex relationship with health necessitate collaborative efforts across disciplines and sectors. Bangladesh's high population density and diverse geographical settings further contribute to the challenges faced in capturing the heterogeneity of health impacts across different regions and population groups. Socioeconomic disparities, cultural variations, and limited access to healthcare further complicate the landscape of conducting research on climate change and health in the country.

Although a national-level initiative is in place to collect real-time healthcare data related to climate changes and other health issues, it has some significant limitations. The major drawback of the current data collection is that while basic patient demographics, such as age and gender, are recorded, vital information such as education, occupation, household wealth, and specific factors contributing to these diseases, remain uncollected. This lack of comprehensive data hampers our understanding of the diseases and our ability to accurately identify high-risk groups, such as age group and educational level. Furthermore, the absence of patients' community characteristics, including place of residence and geographic region, further limits our knowledge of areas prone to climate-sensitive diseases. This may lead to an overrepresentation of disease prevalence in certain districts and divisional facilities where healthcare facilities are mostly located while neglecting others. To improve the initiative's effectiveness, it is crucial to expand data collection efforts to include a more diverse set of healthcare facilities. Additionally, efforts should be made to gather more detailed patient information, such as education, occupation, and household wealth, to gain a better understanding of the social and economic factors influencing disease prevalence. Moreover, incorporating patients' community characteristics, such as place of residence and geographic region, would enable us to identify specific regions at higher risk for climate-sensitive diseases. This knowledge can aid in targeted interventions and resource allocation to address the health challenges effectively.

Another significant limitation is the absence of unique identification numbers to track the health status of patients. This creates challenges in accurately counting patients, as individuals

may be transferred between healthcare facilities or change facilities entirely, resulting in duplicate counts. As a result, the exact number of patients with climate-sensitive diseases remains largely unknown. Additionally, inadequate coverage is a major issue in the current data reporting system. The data recorded in DHIS 2 only represents a subset of healthcare facilities, leaving a significant portion of facilities at the upazila to divisional level unaccounted for. Furthermore, a considerable number of patients with climate-sensitive diseases seek treatment at non-hospital settings, where only aggregate counts are recorded, lacking individual-level data. To overcome these challenges, there is a pressing need for comprehensive improvements in data collection and reporting systems, particularly at the policy and program level.

This study demonstrates several notable strengths as well as a few limitations. One notable strength of this research lies in its ability to offer a comprehensive understanding of climate change and its adverse health impacts. This is achieved through a systematic review and analysis of data from the government data repository, allowing for a comprehensive exploration of climate-sensitive diseases. Strict quality control measures ensured while collecting this data. Additionally, the utilization of the ICD-10 criteria for classifying these diseases and the investigation of the most prevalent conditions provides valuable insights for policymakers. These findings facilitate evidence-based policymaking and the development of targeted programs to address climate-sensitive diseases in Bangladesh.

However, an important limitation of this study is the inability to provide summarized findings due to the inconsistent nature of the available literature. Quantitative data was recorded from 516 healthcare facilities indicating other healthcare facilities data was not recorded. This indicates a report of partial data that we analysed. However, broader geographical coverage of the hospitals from where data was recorded indicate the results are nationally representative. Moreover, the absence of unique patient identification numbers within the dataset poses challenges in distinguishing individuals reported multiple times across various healthcare facilities. Additionally, the lack of relevant data hampers the ability to assess the likelihood of disease occurrence based on respondents' characteristics. The partial data analysis underscores the challenge in formulating comprehensive policies and programs without a holistic understanding of healthcare trends. Without robust data collection methods and unique patient identifiers, policymakers may struggle to tailor interventions effectively to address emerging health issues. However, despite these limitations, this study still holds significant value in enhancing the understanding of the prevalence of climate-sensitive diseases in Bangladesh and informing appropriate response strategies. To address these limitations, it is crucial to enhance the data collection process by incorporating additional patient characteristics, capturing information on the reasons for disease occurrence, and obtaining community-level data. Furthermore, the implementation of unique identification numbers for accurate patient tracking is essential. Expanding the scope of data collection to include a wider range of healthcare facilities is also imperative. By addressing these limitations and obtaining more precise and comprehensive data on climate-sensitive diseases in Bangladesh, policymakers and researchers can develop evidence-based interventions and formulate effective policies to adapt and mitigate the impact of these diseases on public health.

## Conclusion

The existing studies conducted in Bangladesh have only examined a fraction of the total climate-sensitive diseases that are reported in the government data repository. However, these studies have also failed to yield conclusive findings due to limitations such as small sample sizes and restricted coverage of specific geographical areas. Additionally, while the government data repository covers a wide range of climate-sensitive diseases, there are several

identified issues that render it less usable. These include the absence of basic patient characteristics, which hinders comprehensive analysis, and the lack of individual identification, which increases the possibility of reporting the same patient multiple times. These limitations pose challenges for the country in developing evidence-based policies and programs related to climate-sensitive diseases. Given the escalating and ongoing concerns regarding this issue, it is crucial to place greater emphasis on data collection, data analytics and available research. Improving the digital information management system, establishing a centralized database with unique patient identifiers, and providing training for healthcare professionals are essential steps. Integrating data from meteorological agencies into surveillance systems ensures specificity and facilitates the formulation of relevant policies and programs.

## Supporting information

**S1 File. Full list of climatic events.**
(DOCX)

**S2 File. Climate-sensitive disease repository.**
(XLSX)

## Acknowledgments

We acknowledge the support of Climate Change and Health Promotion Unit of the Ministry of Health and Family Welfare, Government of Bangladesh, where this study was conducted and UNICEF Bangladesh for partial funding.

## Author contributions

**Conceptualization:** Md Iqbal Kabir, Md. Nuruzzaman Khan.

**Data curation:** Md Iqbal Kabir, Dewan Mashrur Hossain, Md. Toufiq Hassan Shawon, Md. Mostaured Ali Khan, Md Saiful Islam, As Saba Hossain, Md. Nuruzzaman Khan.

**Formal analysis:** Md. Mostaured Ali Khan, Md. Nuruzzaman Khan.

**Investigation:** Md Saiful Islam, As Saba Hossain.

**Methodology:** Md Iqbal Kabir, Md. Mostaured Ali Khan.

**Resources:** Md. Mostaured Ali Khan.

**Supervision:** Md Iqbal Kabir, Md. Nuruzzaman Khan.

**Validation:** Md Saiful Islam.

**Writing – original draft:** Md. Nuruzzaman Khan.

**Writing – review & editing:** Md Iqbal Kabir, Dewan Mashrur Hossain, Md. Toufiq Hassan Shawon, Md. Mostaured Ali Khan, As Saba Hossain, Md. Nuruzzaman Khan.

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
