## [Decision Letter · Decision Letter 0]

14 Feb 2024

PONE-D-23-35790Addressing Data Challenges for Understanding Climate-Sensitive Diseases in Bangladesh: Evidence from Systematic Review and Government Data RepositoryPLOS ONE

Dear Dr. Kabir,

Thank you for submitting your manuscript to PLOS ONE. After careful consideration, we feel that it has merit but does not fully meet PLOS ONE’s publication criteria as it currently stands. Therefore, we invite you to submit a revised version of the manuscript that addresses the points raised during the review process.

We look forward to receiving your revised manuscript.

Kind regards,

Mohammad Nayeem Hasan

Academic Editor

PLOS ONE

4. We note that Figure 2 in your submission contain [map/satellite] images which may be copyrighted. All PLOS content is published under the Creative Commons Attribution License (CC BY 4.0), which means that the manuscript, images, and Supporting Information files will be freely available online, and any third party is permitted to access, download, copy, distribute, and use these materials in any way, even commercially, with proper attribution. For these reasons, we cannot publish previously copyrighted maps or satellite images created using proprietary data, such as Google software (Google Maps, Street View, and Earth). For more information, see our copyright guidelines: http://journals.plos.org/plosone/s/licenses-and-copyright.

Reviewers' comments:

Reviewer's Responses to Questions

**Comments to the Author**

1. Is the manuscript technically sound, and do the data support the conclusions?

Reviewer #1: Yes

Reviewer #2: Partly

Reviewer #3: Yes

Reviewer #4: Yes

Reviewer #5: Yes

2. Has the statistical analysis been performed appropriately and rigorously? 

Reviewer #1: Yes

Reviewer #2: No

Reviewer #3: Yes

Reviewer #4: Yes

Reviewer #5: N/A

3. Have the authors made all data underlying the findings in their manuscript fully available?

Reviewer #1: Yes

Reviewer #2: Yes

Reviewer #3: Yes

Reviewer #4: Yes

Reviewer #5: Yes

4. Is the manuscript presented in an intelligible fashion and written in standard English?

Reviewer #1: Yes

Reviewer #2: Yes

Reviewer #3: No

Reviewer #4: Yes

Reviewer #5: Yes

5. Review Comments to the Author

Reviewer #1: 1. More details could have been provided on the exact search terms/syntax used in literature review databases to enable replicability. The current supplementary tables do not seem to capture full search strategies.

2. For government data, the quality control and validation measures for data collection/recording could have been elaborately discussed to instill more confidence in findings.

3. The results try to provide a year-wise breakdown of climate sensitive diseases but trends over time are not clearly analyzed through some visual plots or statistical tests. This could have brought out insights.

4. Conclusion seems to predominantly focus on data limitations. Could have provided some specific recommendations on how surveillance and reporting for climate sensitive diseases can be strengthened.

5. The rationale for choice of time period for literature review starting from 2000 needs to be justified.

6. Were there any quality assessment criteria set for including studies in review? This is important to mention.

7. Supplementary Table 13 shows a list of climate-sensitive diseases but it does not match with Figure 3 classifications. Needs consistency.

8. For government data, the sampling framework should be clearly described - how were the 516 facilities selected? Any region wise stratification done? This has implications for results

Reviewer #2: Thanks for your contribution, in short you can modify the title, result part need to include data, analysis need to in good shape like regression analysis, need to add few more figures, discussion part not that much attractive so there is opportunity to work on it and finally a lucrative conclusion along with recommendation is needed.

Reviewer #3: Addressing Data Challenges for Understanding Climate-Sensitive Diseases in Bangladesh: Evidence from Systematic Review and Government Data Repository

The topic is very interesting, and deserve to be studied. However, there are some problems the authors should make them clear before accepting it for publication. The authors please take the following into account.

• The abstract provides a well-organized overview of the study, highlighting the critical issue of understanding climate change-related health outcomes in Bangladesh. While the key components of the research, such as the systematic review and comparison with government data, are effectively communicated, a bit more detail on the specific climate-sensitive diseases and potential implications for policy formulation would enhance the clarity and impact of the abstract. Additionally, ensuring consistency in terminology and briefly explaining key concepts for readers less familiar with the field would contribute to better overall comprehension.

• Consider ensuring a smooth transition between paragraphs, providing a seamless flow of information from the global context to Bangladesh's specific challenges.

• It would be beneficial to briefly list or elaborate on the specific climate-sensitive diseases mentioned in the context of Bangladesh, providing readers with a clearer understanding of the health risks.

• While the mention of the government's Climate Change Strategy and Action Plan is crucial, consider providing a bit more detail on specific measures or strategies outlined in the plan, highlighting how it addresses the health impact of climate change.

• Given that the search was initially conducted in December 2022 and later updated in July 2023, briefly discuss any potential reasons for the update and how it contributed to the study's comprehensiveness.

• Briefly discuss the rationale behind choosing narrative synthesis as the method for summarizing findings, and highlight any key themes or patterns that emerged during this synthesis.

• Author need to check some typesetting errors throughout the manuscript.

• Discuss in more detail the challenges associated with the current data collection through DHIS 2. Highlight specific issues related to patient demographics, community characteristics, and the lack of unique identification numbers.

• Elaborate on the identified limitations of the government-recorded data, emphasizing their impact on policymakers and program developers. Discuss how these limitations may hinder the effective management of climate-sensitive diseases.

• Reinforce the discussion's connection to the study's objectives. Explicitly state how each finding contributes to achieving the research goals, providing a clear linkage between the results and the initial study objectives.

Reviewer #4: Authors conduct a research on “Addressing Data Challenges for Understanding Climate-Sensitive Diseases in Bangladesh: Evidence from Systematic Review and Government Data Repository”. The present manuscript can be accepted for publication in Plos One after addressing the following minor issues.

1. Please reframe the abstract part, current abstract part is not attractive for readers.

2. Please fine tuning the key words

3. Modified the conclusion part for attraction of readers.

4. Update the all references

Reviewer #5: Very nice work on an important issue. I would suggest staying away from bombastic articles from the internet when citing impact of the climate changes (as cited on the page 3, reference 7). Good luck.

6. PLOS authors have the option to publish the peer review history of their article (what does this mean? ). If published, this will include your full peer review and any attached files.

**Do you want your identity to be public for this peer review?** For information about this choice, including consent withdrawal, please see our Privacy Policy .

Reviewer #1: No

Reviewer #2: No

Reviewer #3: No

Reviewer #4: No

Reviewer #5: No

---

## [Author Response · Author response to Decision Letter 1]

1 Jul 2024

We have added a separate MS word file where we provided point-by-point response to each of the reviewers' comments.

---

## [Decision Letter · Decision Letter 1]

16 Sep 2024

PONE-D-23-35790R1Understanding Climate-Sensitive Diseases in Bangladesh using Systematic Review and Government Data RepositoryPLOS ONE

Dear Dr. Kabir,

Thank you for submitting your manuscript to PLOS ONE. After careful consideration, we feel that it has merit but does not fully meet PLOS ONE’s publication criteria as it currently stands. Therefore, we invite you to submit a revised version of the manuscript that addresses the points raised during the review process.

We look forward to receiving your revised manuscript.

Kind regards,

Md Jamal Uddin, Ph.D

Academic Editor

PLOS ONE

**Journal Requirements:**

Reviewers' comments:

Reviewer's Responses to Questions

**Comments to the Author**

1. If the authors have adequately addressed your comments raised in a previous round of review and you feel that this manuscript is now acceptable for publication, you may indicate that here to bypass the “Comments to the Author” section, enter your conflict of interest statement in the “Confidential to Editor” section, and submit your "Accept" recommendation.

Reviewer #2: All comments have been addressed

Reviewer #6: (No Response)

2. Is the manuscript technically sound, and do the data support the conclusions?

Reviewer #2: Yes

Reviewer #6: Yes

3. Has the statistical analysis been performed appropriately and rigorously? 

Reviewer #2: Yes

Reviewer #6: Yes

4. Have the authors made all data underlying the findings in their manuscript fully available?

Reviewer #2: Yes

Reviewer #6: Yes

5. Is the manuscript presented in an intelligible fashion and written in standard English?

Reviewer #2: Yes

Reviewer #6: Yes

6. Review Comments to the Author

**Reviewer #2: ** please give a look again if any major mistake. Climate and its affect is a sensitive issues globally, before final task check it very carefully.

**Reviewer #6:**  1. Nothing was explained about the keyword "District health information system 2" in the abstract section.

2. Is the word "Government data respiratory" correct or should it be "Government data repository" ? Please check throughout the manuscript.

3. Consistency can be maintained with full form and abbreviations. For example no abbreviation is given for Sustainable Development Goals.

4. The keywords that were used for systematic review can be shown in italic format.

5. Please check the title "Exploration of respiratory health data from governmental database" with the word respiratory.

6. The line in the PRISMA description "Ultimately, 70 articles were included in this review, of which 36 were discarded reviewing the study design and finally 34 studies were included in qualitative synthesis. " is somewhat does not match with the diagram. Another box need be added in the diagram where the 70 articles can be added and then 36 exclusion can be shown.

7. By following the standard approach of PRISMA diagram, "identification", "screening" and "included" terms need to be added at the left side of the PRISMA diagram.

8. Please check the typos such as along wise (along with), Randomized control trial (randomized control trial).

9. Some terms need further clarification as it can be hard to understand what they mean. e.g. DALY.

10. No full form is given for MoHFW.

7. PLOS authors have the option to publish the peer review history of their article (what does this mean? ). If published, this will include your full peer review and any attached files.

**Do you want your identity to be public for this peer review?** For information about this choice, including consent withdrawal, please see our Privacy Policy .

Reviewer #2: **Yes: ** Kazi Rakibul Islam

Reviewer #6: **Yes: ** Dr. NAHID SULTANA

---

## [Author Response · Author response to Decision Letter 2]

17 Sep 2024

We have added a MS word file containing a point-by-point response to each of the reviewers' comments.

---

## [Editor Report · Decision Letter 2]

31 Oct 2024

PONE-D-23-35790R2

Understanding Climate-Sensitive Diseases in Bangladesh using Systematic Review and Government Data Repository

PLOS ONE

Dear Dr. Kabir, Thank you for submitting your revised manuscript to PLOS ONE, and for responding to our recent requests regarding your submission. In our editorial checks of the documents that you supplied, we have concluded that your submission does not comply with our policies around data availability. We are therefore rejecting this manuscript.   PLOS journals require authors to make all data necessary to replicate their study’s findings publicly available without restriction at the time of publication (https://journals.plos.org/plosone/s/data-availability). In this case, the following underlying data were not provided as requested: A numbered table of all studies identified in the literature search, including those that were excluded from the analyses.   As a result of these concerns, we cannot consider the manuscript for publication. I am very sorry that we cannot be more positive on this occasion.   Kind regards,

Jennifer Tucker, PhD

Staff Editor

PLOS ONE
---

## [Author Response · Author response to Decision Letter 3]

3 Dec 2024

I have uploaded all details in the MS word files. Please have a look.

---

## [Decision Letter · Decision Letter 3]

6 Feb 2025

Understanding Climate-Sensitive Diseases in Bangladesh using Systematic Review and Government Data Repository

PONE-D-23-35790R3

Dear Dr. Kabir,

We’re pleased to inform you that your manuscript has been judged scientifically suitable for publication and will be formally accepted for publication once it meets all outstanding technical requirements.

Kind regards,

Rajib Chowdhury, M.Sc.; MPH

Academic Editor

PLOS ONE

Additional Editor Comments (optional):

Reviewers' comments:

Reviewer's Responses to Questions

**Comments to the Author**

1. If the authors have adequately addressed your comments raised in a previous round of review and you feel that this manuscript is now acceptable for publication, you may indicate that here to bypass the “Comments to the Author” section, enter your conflict of interest statement in the “Confidential to Editor” section, and submit your "Accept" recommendation.

Reviewer #7: All comments have been addressed

Reviewer #8: All comments have been addressed

2. Is the manuscript technically sound, and do the data support the conclusions?

Reviewer #7: Yes

Reviewer #8: Yes

3. Has the statistical analysis been performed appropriately and rigorously? 

Reviewer #7: Yes

Reviewer #8: Yes

4. Have the authors made all data underlying the findings in their manuscript fully available?

Reviewer #7: Yes

Reviewer #8: Yes

5. Is the manuscript presented in an intelligible fashion and written in standard English?

Reviewer #7: Yes

Reviewer #8: Yes

6. Review Comments to the Author

Reviewer #7: Dear Authors,

The paper is very interesting and well presented now. You have described in detail in the method section how the review was conducted. My suggestion is to further analyze your review. Perhaps you can do it in the different manuscript with a meta analysis.

Reviewer #8: Authors have addressed all the required queries. I am happy to say that Now, it can be accepted for publication.

7. PLOS authors have the option to publish the peer review history of their article (what does this mean? ). If published, this will include your full peer review and any attached files.

**Do you want your identity to be public for this peer review?** For information about this choice, including consent withdrawal, please see our Privacy Policy .

Reviewer #7: No

Reviewer #8: **Yes: ** Abu Reza Md Towfiqul Islam, PhD

---

## [Editor Report · Acceptance letter]

PONE-D-23-35790R3

PLOS ONE

Dear Dr. Kabir,

I'm pleased to inform you that your manuscript has been deemed suitable for publication in PLOS ONE. Congratulations! Your manuscript is now being handed over to our production team.

Kind regards,

on behalf of

Dr. Rajib Chowdhury

Academic Editor

PLOS ONE